# Activation of CD8^+^ T Cell Responses after Melanoma Antigen Targeting to CD169^+^ Antigen Presenting Cells in Mice and Humans

**DOI:** 10.3390/cancers11020183

**Published:** 2019-02-05

**Authors:** Dieke van Dinther, Miguel Lopez Venegas, Henrike Veninga, Katarzyna Olesek, Leoni Hoogterp, Mirjam Revet, Martino Ambrosini, Hakan Kalay, Johannes Stöckl, Yvette van Kooyk, Joke M. M. den Haan

**Affiliations:** 1Amsterdam UMC, Vrije Universiteit Amsterdam, Department of Molecular Cell Biology and Immunology, Cancer Center Amsterdam, Amsterdam Infection and Immunity Institute, De Boelelaan 1117, 1081HV Amsterdam, The Netherlands; d.vandinther@vumc.nl (D.v.D.); m.lopezvenegas@vumc.nl (M.L.V.); henrikeveninga@hotmail.com (H.V.); k.olesek@vumc.nl (K.O.); l.hoogterp@vumc.nl (L.H.); mirjamrevet@hotmail.com (M.R.); m.ambrosini@vumc.nl (M.A.); H.Kalay@vumc.nl (H.K.); y.vankooyk@vumc.nl (Y.v.K.); 2DC4U B.V., De Corridor 21E, 3621 ZA Breukelen, The Netherlands; 3Institute of Immunology, Center of Pathophysiology, Immunology and Infectiology, Medical University of Vienna, 1090 Vienna, Austria; johannes.stoeckl@meduniwien.ac.at

**Keywords:** macrophage, dendritic cell, cancer vaccines, melanoma, T cell responses, CD169, Siglec-1, sialoadhesin

## Abstract

The lack of tumor-reactive T cells is one reason why immune checkpoint inhibitor therapies still fail in a significant proportion of melanoma patients. A vaccination that induces melanoma-specific T cells could potentially enhance the efficacy of immune checkpoint inhibitors. Here, we describe a vaccination strategy in which melanoma antigens are targeted to mouse and human CD169 and thereby induce strong melanoma antigen-specific T cell responses. CD169 is a sialic acid receptor expressed on a subset of mouse splenic macrophages that captures antigen from the blood and transfers it to dendritic cells (DCs). In human and mouse spleen, we detected CD169^+^ cells at an equivalent location using immunofluorescence microscopy. Immunization with melanoma antigens conjugated to antibodies (Abs) specific for mouse CD169 efficiently induced gp100 and Trp2-specific T cell responses in mice. In HLA-A2.1 transgenic mice targeting of the human MART-1 peptide to CD169 induced strong MART-1-specific HLA-A2.1-restricted T cell responses. Human gp100 peptide conjugated to Abs specific for human CD169 bound to CD169-expressing monocyte-derived DCs (MoDCs) and resulted in activation of gp100-specific T cells. Together, these data indicate that Ab-mediated antigen targeting to CD169 is a potential strategy for the induction of melanoma-specific T cell responses in mice and in humans.

## 1. Introduction

Early stage melanoma can adequately be treated with surgical removal of the tumor, however this strategy does not benefit patients with metastatic melanoma. Fortunately, new therapies have greatly improved the prognosis for metastatic melanoma patients over the past years [1]. Two simultaneously approved therapies contributed to the improved survival rates in metastatic melanoma patients. These therapies include BRAF/MEK inhibitors and the immune checkpoint inhibitors anti-programmed cell death protein 1 (PD-1) and anti-cytotoxic T lymphocyte associated protein 4 (CTLA-4) [2,3]. The acknowledgement of cancer immunotherapy as breakthrough of the year in 2013 and the award of the Nobel Prize 2018 for the discoveries of anti-CTLA4 and anti-PD-1 as cancer immunotherapy illustrate the paradigm shift in the treatment of cancer patients [4,5]. Nevertheless, there is still a significant amount of cancer patients that do not benefit from immunotherapy and new strategies should be explored to further improve treatment of melanoma and other cancer patients [6].

Analysis of the tumor microenvironment has improved our understanding of patient responses to immune checkpoint inhibitor immunotherapies [7]. Tumors can be divided into “hot” T cell inflamed and “cold” T cell non-inflamed tumors. Even in “hot” tumors, the tumor micro environment is generally immunosuppressive, which leads to an abundance of exhausted tumor infiltrating CD8^+^ T cells incapable of eliminating the tumor [7,8]. The presence of these pre-existing T cells in the tumor is required for immune blockade inhibitors to induce tumor regression [9,10,11].

One strategy to overcome the immune suppression by the tumor microenvironment is to achieve higher numbers of tumor specific T cells and exploits the specific capacities of dendritic cells (DCs) to induce de novo T cell responses. Initially, the majority of DC vaccines used monocytes or CD34^+^ progenitors that were cultured in vitro to differentiate into DCs. More recently specific DC subsets directly isolated from the blood have been tested [12,13,14]. Although vaccination with ex vivo cultured DCs have proven safe and functional, they have not shown very strong clinical responses [15]. Hence, although promising, there is room for improvement for these personalized treatments which are laborious, time-consuming and require sufficient patient material, specific tools and highly-trained personnel to be successful.

To circumvent the laborious ex vivo DC vaccination therapy, DCs can also be targeted directly in vivo with specific antibodies (Abs) or ligands [16]. Both for ex vivo DC vaccination and in vivo targeting to DCs the selection of the optimal DC subset is important [17]. Classical cDC1 DCs are specialized in cross-presentation of cell-associated antigen (Ag) to CD8^+^ T cells and are identified by DNGR-1/CLEC9a, XCR1, CD8α or CD103 expression in mice and by DNGR-1/CLEC9a, XCR1 and CD141 expression in humans [18,19,20,21,22,23]. The cDC1s contribute to anti-tumor immunity in a plethora of tumor settings in humans and mice including DC vaccination strategies [24].

Pre-clinical studies have shown successful induction of de novo tumor-specific T cell responses after Ab targeting of tumor Ag to different receptors on APCs (reviewed by [25,26]). However, clinical studies remain sparse. DEC205 targeting was the first to be described to induce tumor-specific T cells in mice and more recently DEC205-specific humanized Abs were used in a clinical trial resulting in both cellular and humoral immune responses [27,28]. However, expression of DEC205 in humans is not restricted to DCs, so other receptors might provide better results [29].

We have previously described an alternative in vivo targeting strategy for the activation of CD8^+^ T cell responses in mice [30,31]. High expression of CD169, also known as sialoadhesin and Siglec-1, identifies a unique subset of macrophages in the marginal zone of the spleen and the subcapsular sinus in the lymph node with a privileged location for uptake of Ags (reviewed by [32,33,34,35]). In previous studies, we have shown that ovalbumin (OVA) targeted to CD169 induces T cell responses that rely on Batf3-dependent cDC1s [30,36]. Ag is transferred from CD169^+^ macrophages to cDC1s in the spleen which is required for the induction of T cell immunity. CD8^+^ T cell responses induced via Ag targeting to CD169 inhibit the outgrowth of OVA-expressing tumor cells when either OVA protein or only peptide is used [31], indicating that a single MHC class I epitope can be sufficient for the induction of anti-tumor T cell immunity.

In this study we have further explored the application of CD169 as an in vivo targeting receptor to stimulate melanoma Ag-specific T cell responses in mice and humans. We show that immunization with melanoma peptides conjugated to anti-CD169-Abs results in the induction of melanoma-specific T cell responses in wild type and HLA-A2.1 transgenic mice. Subsequent analysis of CD169 expression in human spleens indicated the presence of CD169/DC-SIGN expressing cells. Finally, in vitro assays with human CD169-expressing MoDCs showed the potential to stimulate human CD8^+^ T cell responses by targeting Ag to human CD169-expressing cells. Together, our data provide further support for the possible use of Ag targeting to CD169 as a therapeutic vaccination strategy against melanoma in humans.

## 2. Results

### 2.1. Targeting Melanoma Ag to CD169 Results in Melanoma Ag-specific T Cell Responses in Mice

Our group has previously used the model Ag OVA to show that peptide targeting to CD169 works as efficiently as protein targeting in a therapeutic vaccination setting against OVA-expressing tumors [31]. To test whether this vaccination strategy is also effective for melanoma-associated Ags, tyrosine-related protein 1 (Trp1), Trp2, gp100 and MelanA recognized by T cells 1 (MART-1-) derived peptides were conjugated to Abs specific for mouse CD169 (MOMA-1), DEC205 (NLDC-145) or an isotype control (R7D4). As most pioneering work with Ag targeting to DCs has been performed by targeting to DEC205 [27], we included DEC205 conjugates as positive controls. We used the H2-K^b^-restricted Trp2_180–188_ epitope (SVYDFFVWL) [37], the H2-D^b^-restricted human gp100_25–33_ epitope (KVPRNQDWL), and a H2-K^b^-restricted modified Trp1_222–229_ epitope (TAYRYHLL) to break tolerance and to induce T cell responses in mice as previously described [38,39]. All peptides contained a C-6-ahx-K (biotin) sequence at the c-terminus and were conjugated as previously described [31,39]. On average, two peptides were conjugated per Ab (Appendix A) and the binding capacity of the Abs after conjugation was confirmed with microscopy or flow cytometry (Appendix A).

For the induction of Trp2-specific CD8^+^ T cell responses in vivo, mice were injected with 1 µg Ab:Ag conjugates in the presence of activating anti-CD40 Ab and Poly(I:C) as adjuvant. 7 days later, the induction of Trp2-specific CD8^+^ T cells was measured by detection of intracellular IFN**γ** production by CD8^+^ T cells after in vitro re-stimulation with the Trp2 peptide (Figure 1A). Targeting to CD169 resulted in clear CD8^+^ T cell responses. Targeting to DEC205 resulted in much lower CD8^+^ T cell responses, while immunization with the isotype control Ab resulted in background level CD8^+^ T cell responses (Figure 1A). The low CD8^+^ T cell responses after Trp2 targeting to DEC205 was unexpected, because it is in contrast to a previous study [40]. However, this can potentially be explained by different administration routes employed (subcutaneous versus intravenous). Next, we immunized with human gp100 and Trp1 conjugates. Immunization with human gp100 resulted in the induction of human gp100-specific CD8^+^ T cells as detected by intracellular IFNγ production (Figure 1B). In addition, cross-reactive mouse gp100-specific T cells, as detected by IFNγ production (Figure 1C) and tetramer binding (Figure 1D) were observed.

In contrast, the targeting with Trp1 conjugates did not induce strong T cell responses specific for Trp1 (Appendix A) which is also reported by others [41]. As shown in Figure 1, the gp100 and Trp-2-specific CD8^+^ T cells produced IL-2 in addition to IFNγ after re-stimulation, which has been shown to be important for CD8^+^ T cell memory induction [42,43]. Taken together, we show that targeting melanoma Ags to CD169 induces multi-functional, Ag-specific CD8^+^ T cell responses in mice that were superior to those obtained by targeting to DEC205.

### 2.2. Targeting HLA A2.1-Restricted MelanA Ag to CD169 Results in Ag-Specific T Cell Responses in HLA A2.1 Transgenic Mice

To test whether this vaccination strategy could also induce CD8^+^ T cell responses against Ags presented in human HLA-A2.1 molecules, we conjugated the MelanA recognized by T cells 1 (MART1)_26–35_ peptide (ELAGIGILTV) to the same Abs (Appendix A). HLA-A2.1 transgenic mice [44] were immunized with these conjugates and the induction of T cell responses was measured by intracellular IFNγ production after peptide re-stimulation in vitro and by tetramer binding. Targeting of MART1 Ag to CD169 and DEC205 resulted in strong multi-functional T cell responses in HLA-A2.1-transgenic mice as shown by IFNγ and IL-2 production upon in vitro re-stimulation with MART1 peptide (Figure 2A). Furthermore, using fully human HLA-A2.1-MART1 tetramers, we observed binding of these tetramers to CD8^+^ T cells (Figure 2B). Because mouse CD8 cannot bind to the α3 domain of human HLA-A2.1 molecules, HLA-A2.1 transgenic mice were used that express a hybrid MHC class I gene that consists of the α1 and α2 domain of the human HLA A2.1 gene and the α3 domain of mouse H-2D^d^ (AAD transgenic mice) [44]. The inability of mouse CD8 to bind to human HLA-A2.1 α3 may cause the lower percentages of tetramer^+^ CD8^+^ T cells compared to the percentages of IFNγ producing CD8^+^ T cells. However, both read-outs indicate strong activation of MART1-specific CD8^+^ T cell responses after CD169 targeting in HLA-A2.1-transgenic mice.

### 2.3. CD169 Expression in Mouse and Human Spleen

Due to the intravenous delivery of the Ab-Ag conjugates, we mainly target to splenic CD169^+^ macrophages present in the marginal zone that line the marginal sinus and are in direct contact with the blood [32]. In human spleen, CD169^+^ macrophages have been described to be located surrounding capillary sheaths in the perifollicular areas of the spleen [45]. To determine whether CD169^+^ macrophages can be readily identified in human spleens, we analyzed the expression pattern of CD169 in mouse and human spleen with immunofluorescence microscopy. CD169-expressing cells were present in both species. While mouse CD169^+^ macrophages could be identified surrounding B cell follicles (Figure 3A), clusters of CD169^+^ cells were observed in the perifollicular area in human spleen (Figure 3B) as previously described [46]. Furthermore, CD169 and DC-SIGN, a C-type lectin receptor, exhibited similar expression patterns in the perifollicular area of the spleen (Figure 3C), while the expression of CD169 was not observed on CD163^+^ macrophages, present in the red pulp of the spleen (Figure 3D). Although initially DC-SIGN was discovered as a receptor on MoDCs [47], DC-SIGN expression is also observed on specific macrophages. Here we show co-expression of CD169 and DC-SIGN on splenic perifollicular macrophages (Figure 3E), which suggests that these macrophages have specialized functions different from the CD163-expressing red pulp macrophages and may be involved in uptake of pathogens. Since previous studies have demonstrated the capacity of DC-SIGN on MoDCs to mediate uptake of Ag and stimulation of Ag-specific T cell responses [48,49,50,51], we set out to investigate Ab-targeting to CD169 and compare it to DC-SIGN targeting on MoDCs.

### 2.4. Ab Binding to CD169 and DC-SIGN on MoDCs Induces Uptake and Internalization

MoDCs express high levels of DC-SIGN and DC-SIGN mediated uptake of Ag-containing Ab and liposomes results in Ag presentation to T cells [47,49,51]. Since MoDCs are known to upregulate CD169 expression after treatment with interferon α (IFNα) [52], we considered that IFNα-treated MoDCs would be a suitable in vitro model system for splenic perifollicular macrophages as both cell types co-expressed DC-SIGN and CD169. First, we treated MoDCs for two days with IFNα to increase the expression of CD169 and analyzed several maturation markers. We observed a pronounced upregulation of the expression of CD169, presence of DC-SIGN, but not of other maturation markers after stimulation with IFNα (Figure 4A). This indicated that IFNα treated-MoDCs can be used as an in vitro model to investigate both CD169 and DC-SIGN-targeting.

MoDCs were incubated with Abs specific for human CD169 (7-239), human DC-SIGN (AZND1) or Langerin (10E2) as a negative control to assess their binding and uptake over time [47,53,54]. Incubation of MoDCs with anti-CD169 and DC-SIGN Abs led to specific binding, however, the uptake kinetics were different when the two receptors were compared (Figure 4B). While DC-SIGN binding resulted in a clear increase in Ab uptake, binding to CD169 showed a much more gradual Ab uptake (Figure 4B). The differences between Ab uptake by the CD169 and DC-SIGN receptors could be a result of a difference in internalization rate. To test this, we first incubated MoDCs with anti-CD169 and anti-DC-SIGN Abs in cold conditions, thoroughly washed unbound Ab away and subsequently incubated the MoDCs at 37 °C for different times to stimulate internalization. Next, we detected the remaining surface-bound Abs with fluorescently labelled anti-mouse Abs. With this approach, we were able to detect the remaining membrane bound Abs as an indirect measure for receptor internalization [55]. In line with previous studies, targeting to DC-SIGN induced a fast internalization of the receptor [51], while targeting to CD169 demonstrated only marginal internalization over time (Figure 4C) [55].

### 2.5. Targeting to CD169 Promotes Ag Cross-Presentation by MoDCs to gp100-Specific T-Cells

Next, we conjugated the human HLA-A2.1 restricted gp100_280–288_ peptide epitope (YLEPGPVTA) to Abs specific for CD169, DC-SIGN and for Langerin as a control Ab (Appendix A).

To determine whether binding of CD169 specific Ab:Ag conjugates can result in Ag presentation to gp100-specific T cells, HLA-A2.1 positive MoDCs were incubated with CD169- or DC-SIGN-gp100 conjugates for 3 hours, washed 2 times and co-cultured with gp100_280–288_-specific T-cells [56]. After 24 hours the IFNγ production in the supernatants of these co-cultures was determined. Compared to the isotype control, Ag targeting to MoDCs via CD169 and DC-SIGN Abs showed significantly enhanced T cell activation (Figure 5). These results indicate that Ag targeting to CD169 on human MoDCs leads to Ag cross-presentation and T cell recognition.

## 3. Discussion

The goal of anti-tumor vaccination strategies is to induce strong tumor-specific CD8^+^ T cell responses that are able to overcome the local tolerogenic mechanisms of the tumor and can efficiently kill tumor cells. CD8^+^ T cell-induced destruction of tumor cells may lead to the release of other tumor Ags and subsequently to broadening of the T cell response [4]. Therefore, the primary induction of strong anti-tumor T cell responses is critical [57]. The efficacy of an anti-tumor vaccine will depend on several factors. The vaccine should (1) result in efficient Ag cross-presentation by DCs, (2) include tumor-specific or tumor-associated Ags that can overcome existing CD8^+^ T cell tolerance and (3) contain the right adjuvant to support the induction of strong CD8^+^ T cell responses. Our study has addressed the first point and proposes CD169-targeting as a possible strategy to achieve strong cross-presentation and CD8^+^ T cell activation using mouse models as well as in vitro human models.

Many different forms of vaccines are being investigated for the activation of cancer-specific immune responses [58]. One potential route to obtain strong cross-presentation by DCs is to use Abs or ligands to target Ags to uptake receptors on these cells. Ag targeting to DEC205, CLEC9a, CLEC12a, DC-SIGN, and Mannose Receptor have been shown to induce T cell responses in mice and in human in vitro culture systems [25,26]. However, clinical applications have been limited to DEC205 targeting and the broad expression profile of DEC205 might not be optimal for targeting in patients [29]. This underlines the need for the identification of optimal targeting receptors in humans.

In our previous studies we have investigated the efficacy of Ag targeting to CD169^+^ macrophages in mouse models and demonstrated induction of strong effector and memory CD8^+^ T cell responses and suppression of tumor outgrowth after either OVA protein or SIINFEKL peptide targeting in the presence of a strong adjuvant [31]. In the present study we show that targeting of several melanoma epitopes to CD169^+^ macrophages elicits CD8^+^ T cell responses that were stronger or equivalent to those elicited by DEC205 targeting. CD169-targeting was successful for both H-2K^b^, H-2D^b^ and HLA-A2.1-restricted epitopes. Noteworthy, targeting of Trp2, an epitope entirely shared among mouse and human, resulted in strong CD8^+^ T cell responses, which shows that CD169-targeted vaccination can break tolerance against this epitope. In addition, these results indicate that the capture of Ab:Ag conjugates by CD169^+^ macrophages results in efficient Ag transfer to cross-presenting cDC1s as we have previously described and is not dependent on the specific Ag used [30,36]. While the use of HLA-A2.1 transgenic mice is a first step towards clinical translation, a humanized mouse model with functional human lymphoid tissue including CD169^+^ macrophages and DC subsets could be used to further validate our findings [59].

The microanatomy of the spleen has been described to differ significantly between mice and humans [45,46]. Mouse CD169^+^ macrophages are located in the marginal zone, lining the marginal sinus cells, where they function to capture blood-borne pathogens and to interact with immune cells [35,60]. The marginal zone that delineates B cell follicles and separates the white and red pulp in mice, is absent in human spleens. In contrast, human splenic CD169^+^ macrophages are associated with the endings of capillary sheaths in the perifollicular area, an area which mice lack [45,61]. However, the presence of CD169^+^ macrophages surrounding the open endings of the capillaries, would also enable them to capture blood-borne pathogens before entering the spleen, similar to mouse CD169^+^ macrophages.

Our results demonstrate co-expression of DC-SIGN and CD169 on these perifollicular cells in human spleen. A similar population of CD169^+^ DC-SIGN^+^ cells appears to be present in human lymph node sinuses [62]. Interestingly, CD169 and DC-SIGN are both uptake receptors for pathogens such as HIV, which supports the hypothesis that CD169^+^ DC-SIGN^+^ macrophages have a pathogen scavenging function in humans and in mice [47,52,63,64,65]. The spatial and functional resemblance between these uptake receptors prompted us to test both DC-SIGN and CD169 targeting on human cells in vitro. We used IFNα-treated MoDCs as an in vitro model system for perifollicular macrophages, because they co-express DC-SIGN and CD169 [52]. Our data show that there is uptake and internalization of both receptors upon Ab binding, although DC-SIGN is internalized faster and to a greater extent. Internalization rates and routing differences could potentially depend on the Abs used or the region where they bind to [66,67]. Alternatively, lower internalization rates of human CD169 compared to DC-SIGN might suggest an Ag retention function for human CD169 similar to that observed for mouse CD169 [32,68]. This would contribute to known functions of CD169^+^ macrophages, such as trans-infection of CD169 bound viruses and transfer of intact Ag to B cells [68,69,70]. Together, these data suggest that binding to CD169 would result in longer retention of intact Ags.

We have previously reported that mouse CD169^+^ macrophages preferentially transfer Ag to cDC1 and that these cells are responsible for T cell cross-priming [36]. Whether human CD169^+^ macrophages interact with human cross-presenting cDC1s to transfer Ag and to activate T cell responses is still unknown. However, after Ag targeting to CD169, MoDCs were themselves able to activate CD8^+^ T cells indicating Ag cross-presentation. Cross-presentation by CD169^+^ macrophages has been proposed in mice, but the relevance for the activation of primary CD8^+^ T cell responses is not clear yet [71,72]. In humans a third Ag presenting cell subset might be involved. The recently described Axl^+^Siglec-6^+^ DCs have been described to express CD169 and are capable of T cell activation [73,74,75]. Further analysis of these cells could define their contribution to the induction of CD8^+^ T cell responses after targeting with CD169-specific Ab-peptide conjugates.

Our study has focused on the most efficient method of Ag targeting and CD8^+^ T cell activation and did not investigate which Ags would be most successful to induce clinical responses. Current DC vaccination trials with melanoma patients use different types of tumor-associated Ags for loading of DCs for vaccination such as tumor lysate, tumor cell mRNA and several non-mutated self-Ags (e.g., gp100, MART1, tyrosinase) [16]. These non-mutated self-Ags could be easily integrated into an off-the-shelf vaccine with CD169-specific Abs. As the fast identification of patient-specific neo-Ags has recently become feasible, vaccination with personalized neo-Ags may be used to induce stronger clinical responses [76,77,78,79]. First clinical data (a Phase I clinical trial) using personalized neo-Ag vaccination with DCs have shown broadening of T cell responses in melanoma patients [80]. The vaccination with CD169-specific Abs conjugated to neo-Ag peptides would be an alternative approach.

## 4. Materials and Methods

### 4.1. Mice

C57Bl6/J, and B6.Cg(CB)-Tg (HLA-A/H2-D)2Enge/J mice stock #004191 [44] were obtained from Charles River (Den Bosch, the Netherlands) and the Jackson Laboratory (Bar Harbor, Maine, USA), respectively and bred at the animal facility of the Amsterdam UMC, Vrije Universiteit Amsterdam. Mice used in this study were kept under specific pathogen-free conditions and were age/sex matched. Experiments were approved by the Dutch National Animal Ethics Committee “dierexperimentencommissie” or the “centrale commissie dierproeven” (Approved on 10 April 2017, CCD protocol AVD1140020171024).

### 4.2. Peptide Synthesis

The antigenic peptides hgp100_(25–33)_ (KVPRNQDWL and KVPRNQDWLC-6-ahx-lysine (biotin)) [81], mgp100_25–33_ (EGSRNQDWL), human gp100_280–288_ (YLEPGPVTA and YLEPGPVTAC-6-ahx-lysine (biotin)), Trp1_222–229_ (TAYRYHLL and TAYRYHLLC-6-ahx-lysine (biotin)), Trp2_180–188_ (SVYDFFVWL and SVYDFFVWLC-6-ahx-lysine (biotin)) and MART1_(26–35)_ (ELAGIGILTV and ELAGIGILTVC-6-ahx-lysine (biotin)) were synthesized at the GlycO2peptide unit at our lab by solid phase peptide synthesis using Fmoc chemistry on a Symphony peptide synthesizer (Protein Technologies Inc., Tucson, AZ, USA). The linker 6-ahx and the special amino acid lysine(biotin) were purchased at Iris Biotech GmbH (Marktredwitz, Germany) The peptides were purified on a preparative Ultimate 3000 HPLC system (Thermo Fisher Scientific, Breda, the Netherlands) over a Vydac 218MS1022 C18 25 × 250mm column (Grace Davidson, Worms, Germany). Quality control was performed by UPLC-MS on a Ultimate 3000 UHPLC system (Thermo Fisher Scientific) hyphenated with a LCQ-Deca XP Iontrap ESI mass spectrometer (Thermo Finnigan, Waltham, MA, USA) using a RSLC 120 C18 Acclaim 2.2um particle 2.1 × 250 mm column and ionizing the sample in positive mode.

### 4.3. Ab-Ag Conjugates

hgp100 (KVPRNQDWLC-6-ahx-lysine (biotin)), Trp1 (TAYRYHLLC-6-ahx-lysine (biotin)), Trp2 (SVYDFFVWLC-6-ahx-lysine (biotin)), MART1 (ELAGIGLTVC-6-eahx-lysine (biotin)) peptides were conjugated to purified Abs anti-mCD169 (MOMA-1), anti-mDEC205 (NLDC-145) and a rat IgG2a isotype control (R7D4). For the human studies, gp100 (YLEPGPVTAC-6-ahx-lysine (biotin)) peptide was conjugated to anti-hCD169 (7-239), anti-hDCSIGN (AZN-D1) and the isotype control anti-hLangerin (10E2) [47,53,54]. Conjugation of peptides to Abs was realized via a sulfhydryl based coupling, as described previously [39]. In short, purified Abs were functionalized with 2–8 equivalents of SMCC (succinimidyl 4-(N-maleimidomethyl) cyclohexane-1-carboxylate, Thermo Fisher Scientific) in phosphate buffer pH 8.5. After purification over PD-10 columns (GE Life Sciences, Eindhoven, The Netherlands) against phosphate buffer pH 7.2 activated Abs were concentrated with centricon 30 (Merck Millipore, Amsterdam, The Netherlands) down to 500 µL. 2–4 Equivalents of peptides in 50 µL DMSO was added to the Abs and after 1 h incubation at room temperature conjugates were purified over a Sephadex 75 10/30 column (GE Life Sciences) according to manufacturer’s HPLC settings.

Concentration (Pierce BCA Protein Assay Kit, Thermo Fisher Scientific), amount of biotinylated peptides per Ab (FluoReporter™ Biotin Quantitation Assay Kit, Thermo Fisher Scientific) and endotoxin amount (<0.01 EU/mL with Pierce LAL chromogenic Endotoxin quantication kit, Thermo Fisher Scientific) was determined for all conjugates.

### 4.4. Immunization and In Vitro Restimulation

Mice were intravenously immunized with 1 µg of the indicated Ab:Ag conjugates in the presence of 25 µg anti-CD40 Ab (1C10) and 25 µg Poly(I:C). 7 days after immunization spleens were collected and single cell suspensions were generated using a 100 µm strainer. Red blood cells were lysed with in house ACK buffer (0.15 M NH_4_Cl, 10 mM KHCO_3_, 0.1 mM Na_2_EDTA on H_2_O with pH of 7.2–7.4). 3 × 10^6^ splenocytes were incubated for 5 h at 37 °C with 1 µg/mL of indicated peptides Trp2 (SVYDFFVWL), hgp100 (KVPRNQDWL), mgp100 (EGSRNQDWL) or MART1 (ELAGIGILTV) in 200 µL RPMI (10% FCS, 2% PSG, 50 µM βmercapto-ethanol) in the presence of GolgiPlug according to manufacturer’s protocol (BD Biosciences, Temse, Belgium).

### 4.5. Flow Cytometry

Mouse splenocytes were isolated, cultured and stained after Fc-block with in-house produced 2.4G2 supernatant. Tetramer staining was performed in FACS buffer (0.5% BSA + 0.02% NaN_3_) in the presence of anti-CD8α (clone 53-6.7) for 1 h at 37 °C, subsequent surface marker staining was done 30 min on ice. Tetramers were a kind gift from J.W. Drijfhout (LUMC, Leiden, the Netherlands). After restimulation cells were stained for surface markers in FACS buffer, fixed in 2% PFA (pH 7.2) and intracellular cytokine staining was performed in FACS buffer with 0.5% saponine. Human MoDCs were stained for surface markers in FACS buffer, 30 min on ice and fixed in 2% PFA (pH 7.2). Samples were measured on the Cyan (Beckman Coulter, Brea, CA, USA) or the LSRFortessa (BD Biosciences) and analyzed using FlowJo, LLC (Ashland, Oregon, USA). Abs: anti-mCD8α (clone 53-6.7) eBioscience, anti-mIFNγ (clone xM61.2) eBioscience (Santa Clara, CA, USA), anti-mIL-2 (clone JES6-5H4) eBioscience, anti-mCD44 (HI44A) ImmunoTools, anti-hDC-SIGN (Clone AZND-1), anti-hCD169 (clone 7-239), anti-hLangerin (clone 10E2), Alexa Fluor^®^ 488 labeled goat anti-mouse secondary Abs (Life Technologies, Carlsbad, CA, USA). Fixable viability dye eFluor™ 780 (eBioscience™).

### 4.6. Immunofluorescence

Human spleen tissue was obtained from the VUmc Biobank Pathologie (VUmc 2015-074 approved on 31 August 2015) and collected after approval by the Medical Ethical Committee of the VUmc (Amsterdam, The Netherlands) in accordance with the Declaration of Helsinki and according to Dutch law. Tissues were used anonymously, and no information is available on age or gender of the donors. Mouse and human tissue was cryopreserved in Tissue Tek^®^ (Sakura, Alphen aan de Rijn, The Netherlands), cut at 5–10 µM on a Mikrom HM 560 cryostat and fixed in 100% aceton. Abs were diluted in PBS containing 2% new born calf serum. Pictures were taken at a LEICA DM6000 (Amsterdam, The Netherlands). LAS AF software was used for acquisition, ImageJ for image processing. Abs: anti-mCD169 (clone SER4), anti-IgD (clone 11-26c.2a) Biolegend (San Diego, CA, USA), anti-VCAM/CD106 (clone 429) eBioscience, anti-hCD19 (clone 11G1), anti-hDC-SIGN (clone AZND1), anti-hCD169 (clone 7-239), anti-hCD169-biotin (Miltenyi Biotech, Bergisch Gladbach, Germany), anti-hCD163 (clone EDHU1) Serotec (Bio Rad Abs, Oxford, UK). Alexa Fluor^®^ 488 labeled goat anti-mouse secondary Abs (Life Technologies) and Alexa Fluor^®^ 555 conjugated streptavidin (Invitrogen, Waltham, MA, USA) were used at 5 µg/mL to detect the unlabeled and biotinylated primary Abs, respectively. Unless stated otherwise primary Abs were produced by the Mo2Ab Facility MCBI Amsterdam UMC (Amsterdam, The Netherlands).

### 4.7. Generation of MoDCs

Human blood monocytes were isolated from buffy coats (Sanquin Blood Bank, Amsterdam, The Netherlands) by percoll gradient centrifugation. Immature DCs were generated by culture in RPMI (10% FCS, 1% PSG) supplemented with 12.5 ng/mL of recombinant human IL-4 and GM-CSF (Biosource, ThermoFisher Scientific) for 4 days. For the upregulation of CD169, immature MoDCs were cultured for two days with 1000 IU/mL recombinant human IFN-α 2a (Immunotools, Friesoythe, Germany).

### 4.8. Binding and Uptake Assay

IFNα treated MoDCs were seeded in RPMI medium at a concentration of 2 × 10^6^ cells/mL in 96-U bottom well plates, 5 × 10^4^ cells per well. After precooling the MoDCs at 4 °C for 15 min, cells were incubated with 2 µg/mL of anti-DC-SIGN, anti-CD169 or control Abs at 4 °C for 45 min and after with Alexa Fluor^®^ 488 labeled goat anti-mouse Abs at 4 °C for 45 min. Next, MoDCs were placed on ice or incubated at 37 °C for 15, 30 or 60 min, washed in FACS buffer and fixed in 2% PFA. Binding of the Abs and uptake at different time points was assessed by flow cytometry.

### 4.9. Internalization Assay

IFNα treated MoDCs were seeded at a concentration of 2 × 10^6^ cells/mL in 96-U bottom well plates, 5 × 10^4^ cells per well and pre-incubated at 4 °C during 15 min with 20 µg/mL of anti-hDC-SIGN, anti-hCD169 or control Abs. Next, the cells were thoroughly washed in RPMI medium, to remove excess of unbound Abs, and further incubated either on ice or at 37 °C during 15, 30 or 60 min. Finally, MoDCs were incubated with Alexa Fluor^®^ 488 labeled goat anti-mouse secondary Abs at 4 °C for 45 min, washed in FACS buffer and fixed in 2% PFA. Remaining surface receptors were assessed by flow cytometry.

### 4.10. Ag Presentation

IFNα treated HLA-A2^+^ MoDCs were seeded in RPMI medium at a concentration of 2 × 10^4^ cells per well in 96-U bottom well plates. After a pre-incubation of 15 min at 4 °C, cells were incubated with different concentrations (2.2–20 µg/mL) of anti-hDC-SIGN, anti-hCD169 or control Abs conjugated to the hgp100 peptide (YLEPGPVTA) in RPMI medium supplemented with 100 ng/mL (Sigma) LPS at 4 °C for 45 min. Next, MoDCs were washed in RPMI to remove excess of unbound Abs and incubated at 37 °C. After 3 h, MoDC were co-cultured overnight with a gp100_280–288_ TCR transduced HLA-A2.1 restricted T-cell line [56], at a concentration of 10^5^ cells per well (ratio MoDC:T-cells 1:5). After 24 h, production of IFNγ in the supernatants of the co-cultures was determined by ELISA (Biosource, San Diego, CA, USA).

### 4.11. Statistical Analysis

For the in vivo experiments a one-way ANOVA with Sidak’s multiple comparison test was used. For the human in vitro experiments a matched one-way ANOVA with Sidak’s multiple comparison test was performed using GraphPad Prism 7 (San Diego, CA, USA).

## 5. Conclusions

In conclusion, our data indicate that it is possible to induce melanoma-specific T cell responses in vivo and in human cell culture systems after targeting Ag to CD169 on Ag presenting cells. Because of the presence of CD169^+^ macrophages in human spleen, we propose that CD169-targeting may be feasible and potentially efficient to induce CD8^+^ T cell responses in humans as we have demonstrated in mice. Further studies should explore the application of in vivo targeting of melanoma Ags to human CD169^+^ macrophages to induce clinical responses in combination with immune checkpoint inhibitors.

## Figures and Tables

**Figure 1 cancers-11-00183-f001:**
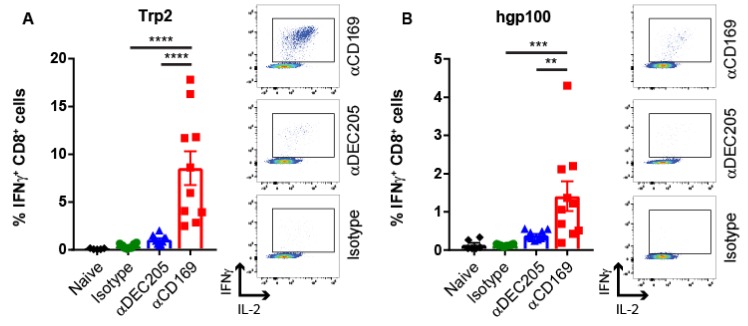
Targeting melanoma Ag to CD169 results in specific T cell responses in mice. Intravenous immunization with 1 µg Ab:Ag conjugates in the presence of 25 µg anti-CD40 Ab and 25 µg Poly(I:C). (**A**) Percentage of IFNγ producing CD8^+^ T cells after 5 h in vitro restimulation with Trp2 peptide 7 days after immunization with Trp2:Ab conjugates. (**B**–**C**) Percentage of IFNγ producing CD8^+^ T cells after 5 h in vitro restimulation with (**B**) human gp100 peptide or (**C**) mouse gp100 peptide 7 days after immunization with human gp100:Ab conjugates. (**D**) Percentage of CD8^+^ T cells binding H2-D^b^ tetramers 7 days after immunization with human gp100:Ab conjugates. Combined data of two experiments with 4–6 mice per group with one representative dotplot of each group is shown for all figures except for C which is one experiment with 6 mice per group. Statistical analysis one-way ANOVA with Sidak’s multiple comparison test * *p* < 0.05, ** *p* < 0.01, *** *p* < 0.001, **** *p* < 0.0001.

**Figure 2 cancers-11-00183-f002:**
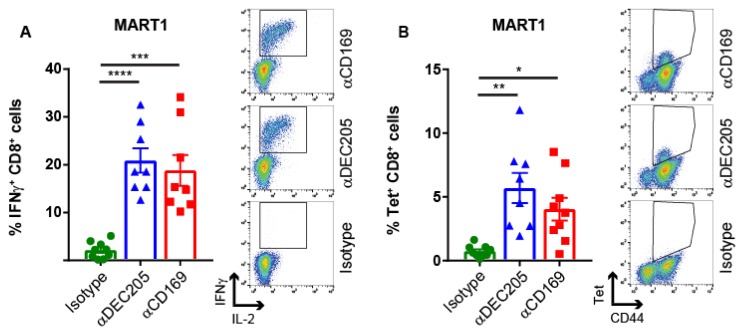
Targeting HLA-A2.1 restricted Ag to CD169 results in Ag-specific T cell responses in HLA A2.1 transgenic mice. Intravenous immunization with 1 µg Ab:Ag conjugates in the presence of 25 µg anti-CD40 Ab and 25 µg Poly(I:C). (**A**) Percentage of IFNγ producing CD8^+^ T cells after 5 h in vitro restimulation with MART-1 peptide (**B**) Percentage of CD8^+^ T cells binding HLA-A2.1 tetramers 7 days after immunization with MART-1:Ab conjugates in HLA-A2.1 transgenic mice. Combined data of two experiments with 3–6 mice per group with one representative dotplot of each group is shown. Statistical analysis one-way ANOVA with Sidak’s multiple comparison test * *p* < 0.05, ** *p* < 0.01, *** *p* < 0.001, **** *p* < 0.0001.

**Figure 3 cancers-11-00183-f003:**
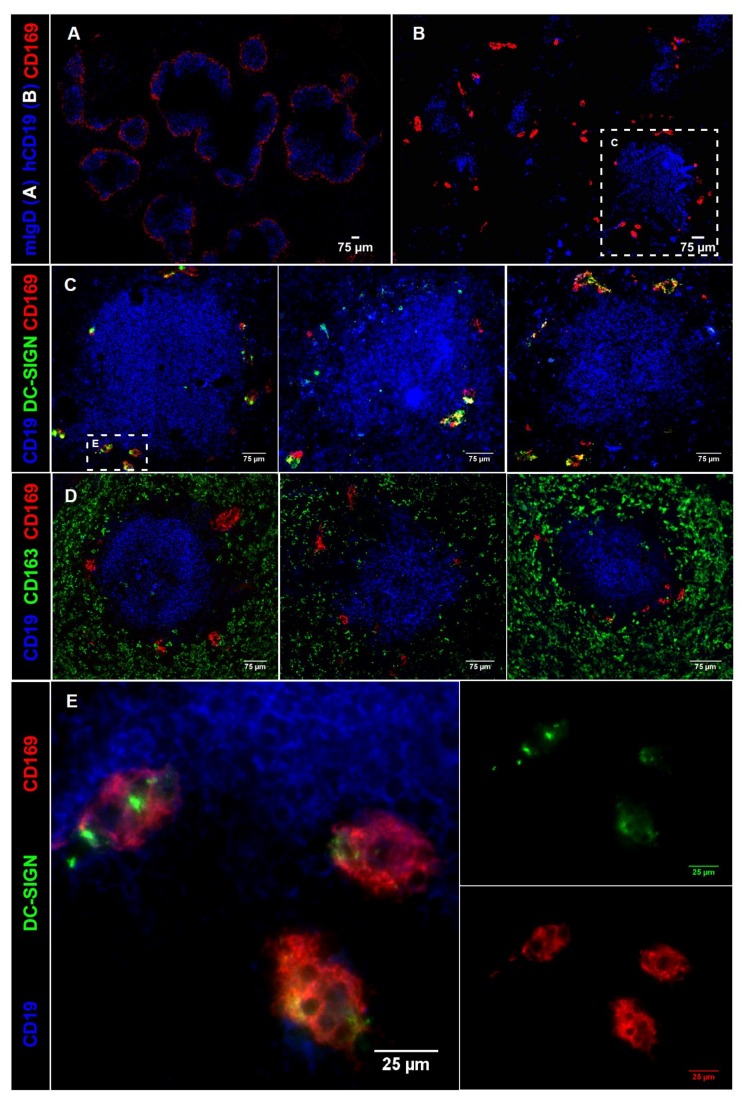
CD169 expression in mouse and human spleen. (**A**) Mouse spleen overview section stained for IgD (blue) and CD169 (red). (**B**) Human spleen overview section stained for CD19 (blue) and CD169 (red). (**C**) B cell follicles (blue) in human spleen from three different donors, CD169^+^ cells (red) and DC-SIGN^+^ cells (green). The first panel is a zoom-in from (**B**). (**D**) B cell follicles (blue) in human spleen from three different donors, CD169^+^ cells (red) and CD163^+^ cells (green). (**E**) zoom-in from first panel of (**C**).

**Figure 4 cancers-11-00183-f004:**
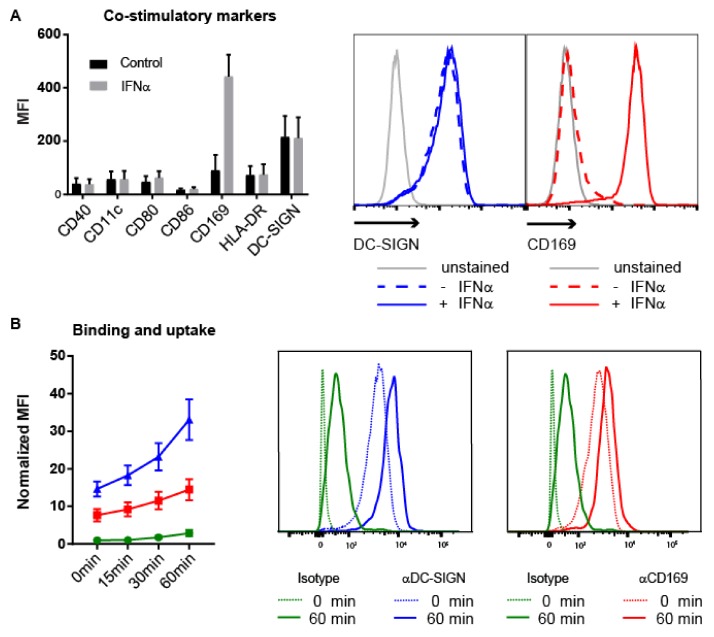
Expression of CD169 and DC-SIGN on human MoDCs and binding and internalization of Ab-Ag conjugates. (**A**) Expression of DC maturation markers and CD169 on unstimulated MoDCs (black) or IFNγ stimulated MoDCs (grey) (*n* = 5). (**B**) Binding and uptake of specific Abs for CD169 (red), DC-SIGN (blue) and an isotype control (green). Cells were incubated with Abs for 45 min on ice, directly followed by 45 min incubation on ice with a secondary Ab. Subsequently, cells were kept on ice (*t* = 0 min), or incubated at 37 °C for indicated times and finally analyzed by flow cytometry (*n* = 4). (**C**) Receptor internalization after incubation with specific Abs for CD169 (red), DC-SIGN (blue) and an isotype control (green). Cells were incubated with the Abs for 45 min on ice and subsequently washed to remove unbound Ab. Next, cells were kept on ice (*t* = 0 min) or incubated at 37 °C for indicated times. Receptors remaining on the cell surface were visualized by flow cytometry after incubation with a secondary Ab (*n* = 3). (**A**–**C**) Representative histograms of DC-SIGN (blue, middle panel) and CD169 (red, right panel) expression are shown. (**B**,**C**) MFIs are normalized to isotype control at 0 min.

**Figure 5 cancers-11-00183-f005:**
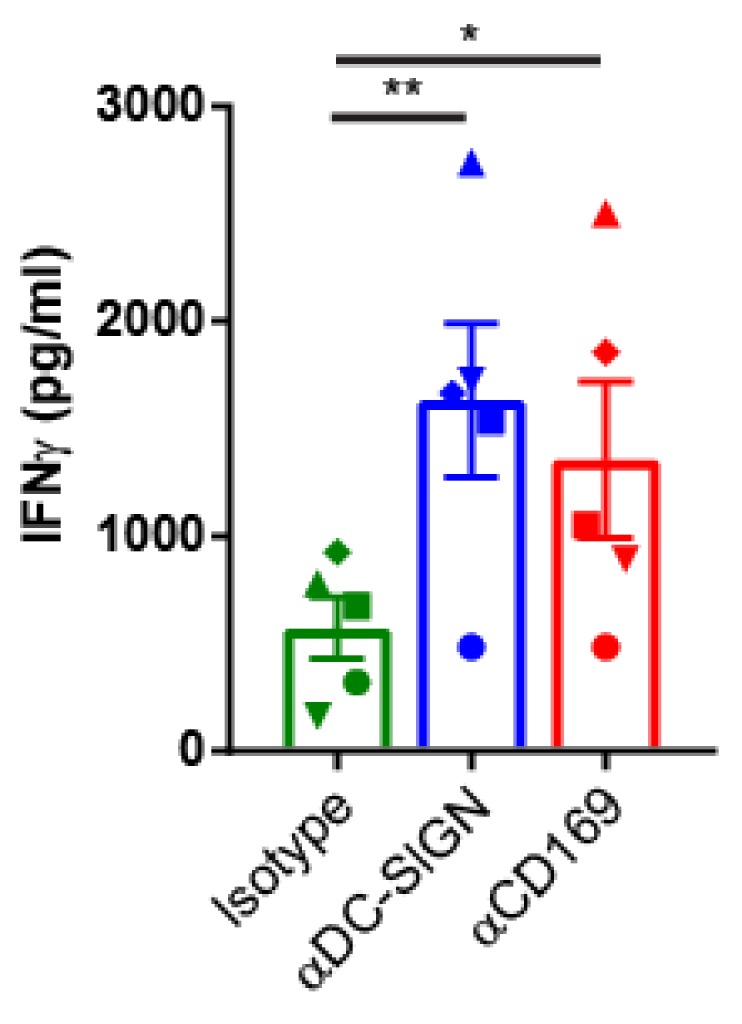
Targeting melanoma Ag to CD169 on human MoDCs results in melanoma Ag presentation to T cells. IFNγ production by gp100_280–288_-specific HLA-A2.1 restricted T cell line after 24 h co-culture with MoDCs previously pulsed for 3 h with Ab:gp100 c onjugates specific for CD169, DC-SIGN or isotype control Ab (*n* = 5 healthy donors, each different symbol represents one donor). Statistical analysis matched one-way ANOVA with Sidak’s multiple comparison correction * *p* < 0.05, ** *p* < 0.01.

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
