# Peer review of "Activation of CD8+ T Cell Responses after Melanoma Antigen Targeting to CD169+ Antigen Presenting Cells in Mice and Humans"

_cancers, 2019, doi:10.3390/cancers11020183_

Round 1
Reviewer 1 Report
The authors extend their previous study on the interest to target CD169 macrophages to improve T cell priming after vaccination. Results are well controlled.
Some suggestions to improve the quality of the manuscript :
- What about the induction of specific CD4+T cells after CD169+ macrophage targeting. It will indicate the ability of this vaccine strategy to allow long term CD8+T cell memory
- The authors compare in vivo antibodies targeting Dec-205 or CD169 and in vitro the efficiency of antibodies to DC-sign or CD69 for cross-presentation. What about the respective efficiency of DC-Sign and CD169 in vivo. In the discussion the respective advantages of these two antibodies could be discussed.
- The authors use very strong adjuvants (anti-CD40 combined with poly (I :C). Does this strategy also work with other adjuvants.
Author Response
The authors extend their previous study on the interest to target CD169 macrophages to improve T cell priming after vaccination. Results are well controlled.
Some suggestions to improve the quality of the manuscript :
We would like to thank the reviewer for the valuable suggestions.
- What about the induction of specific CD4+T cells after CD169+ macrophage targeting. It will indicate the ability of this vaccine strategy to allow long term CD8+T cell memory.
As the reviewer points out, in this manuscript we have only tested the induction of CD8+ T cell responses after targeting with melanoma epitopes. In previous work we have demonstrated the activation of ovalbumin-specific CD4+ T cell responses after ovalbumin targeting to CD169+ macrophages (Veninga et al. Eur. J. Immunol. 2015, 45:747). These CD4+ T cells were essential for the generation of ovalbumin-specific B cell responses.
In another study we have compared targeting of SIINFEKL, the immunodominant CD8 epitope of ovalbumin, and the whole ovalbumin protein to establish the ability of a peptide vaccination strategy to induce long term CD8+ T cell memory (van Dinther et al. Front. Immunol. 2018, 9:1997). We observed very strong effector and memory CD8+ T cell responses after targeting with SIINFEKL, and detected no difference between SIINFEKL and ovalbumin targeting to CD169 with regard to the long term memory or recall CD8+ T cell response. These results indicate that ovalbumin-specific CD4+ T cells are likely not essential for CD8+ T cell memory in our vaccination model, which can be explained by the potent adjuvant used, anti-CD40, as it is known to mimic the help by CD4+ T cells.
We have adjusted the discussion to address this point (page 10 line 266).
- The authors compare in vivo antibodies targeting Dec-205 or CD169 and in vitro the efficiency of antibodies to DC-sign or CD169 for cross-presentation. What about the respective efficiency of DC-Sign and CD169 in vivo. In the discussion the respective advantages of these two antibodies could be discussed.
DEC205 is a well-established receptor for dendritic cell targeting in mice and for this reason we chose to compare mouse CD169 targeting with DEC205 targeting. In contrast, DC-SIGN has multiple homologues in mice and it is not clear which mouse homologue is most representative for DC-SIGN expression and function. Alternatively, different types of transgenic mice have been generated that express human DC-SIGN under the control of different promotors. Although these transgenic mice could be used for vaccinations to compare human DC-SIGN with mouse CD169 targeting, we reasoned that this would not be a valid comparison and chose to compare human DC-SIGN and human CD169 targeting on monocyte-derived dendritic cells.
In our in vitro studies we observed differences in the internalization of DC-SIGN and CD169 antibodies by moDCs. DC-SIGN internalization was faster than CD169 internalization and this could have implications for antigen retention, degradation, transfer to dendritic cells and presentation. We have discussed these aspects on page 10 line 297- page 11 line 315. Since it is not clear whether direct presentation by CD169 expressing antigen presenting cells or antigen transfer to classical dendritic cells will be more important in the human situation, we did not further speculate whether DC-SIGN or CD169 targeting will be more efficient in humans.
- The authors use very strong adjuvants (anti-CD40 combined with poly (I :C). Does this strategy also work with other adjuvants.
The reviewer comments on an important factor in our vaccination strategy. Anti-CD40 and Poly(I:C) are indeed strong adjuvants. Agonistic anti-CD40 antibodies, such as used in this study, are currently being tested in clinical trials. In the case that anti-CD40 therapies would be approved for the treatment of cancer, it might be beneficial to combine these with our vaccination strategy. As the reviewer points out, other adjuvants should be considered and tested in pre-clinical and clinical settings. Unfortunately, we have yet not been able to address this and we believe that this is beyond the scope of the study presented here.

Reviewer 2 Report
This manuscript focuses on validating an antibody targeting melanoma peptides to CD169+ macrophages and human monocytes to elicit melanoma-specific CD8+ T cell responses. I consider this work potentially interesting, however there are some major weaknesses that need to be addressed.
First, there is significant overlap with recent papers published this year by the same group. They first described this CD169-targeting antibody to deliver viral and model antigens to CD169+ macrophages and subsequently to cross-presenting DCs and initiate CD8+ T cell responses. They next validated that this strategy can be harnessed to fight melanoma tumors using the model antigen ovalbumin. Hence the novelty of this manuscript relies in the validation of this same concept using more relevant melanoma antigens. However, this validation falls short of some aspects.
The selection of human peptides to break tolerance against mouse epitopes seems not the best option unless they are planning to validate this strategy for xenogeneic vaccines, which are not necessarily translatable to a clinical setting. It would be far more important to put the focus on epitopes that are entirely shared among mouse and human amino acidic sequences. In this way you validate the CD169 Ab in a system that incorporates the issue of breaking tolerance against self-melanoma antigens. Actually, the peptide Trp2(180-188) tested by the authors fulfills this criterion. Indeed, this peptide has been also described as an HLA-A2-restricted epitope, which is ideal to test in both wild-type H-2b and transgenic HLA-A2 mice.
To really validate anti-tumor responses generated by the anti CD169 antibody linked to melanoma peptides, it would be needed to test this against a melanoma tumor model, such as wild type and HLA-A2-expressing B16 melanoma in a therapeutic or at least preventive settings.
In addition, to this reviewer it is not clear what is the degree of novelty of figure 3. It seems that all the data presented have been previously shown. What is the contribution of this figure to the manuscript?
In the same line, there is no clear connection of the main topic of the manuscript with the analysis of the binding, uptake and internalization of the anti-CD169 Ab in human monocytes. The characterization of this antibody has been made after in vivo i.v. delivery, where it is uptaken by CD169 macrophages and then by DCs. However, it is not clear why is important to study this in human monocytes. It could be relevant to study this if the anti CD169 antibody was planned to be used in ex vivo monocyte-derived vaccination protocols. In that case, what would be the advantages of using this antibody?
Author Response
This manuscript focuses on validating an antibody targeting melanoma peptides to CD169+ macrophages and human monocytes to elicit melanoma-specific CD8+ T cell responses. I consider this work potentially interesting, however there are some major weaknesses that need to be addressed.
First, there is significant overlap with recent papers published this year by the same group. They first described this CD169-targeting antibody to deliver viral and model antigens to CD169+ macrophages and subsequently to cross-presenting DCs and initiate CD8+ T cell responses. They next validated that this strategy can be harnessed to fight melanoma tumors using the model antigen ovalbumin. Hence the novelty of this manuscript relies in the validation of this same concept using more relevant melanoma antigens. However, this validation falls short of some aspects.
We would like to thank the reviewer for the comments and feedback on our manuscript.
The selection of human peptides to break tolerance against mouse epitopes seems not the best option unless they are planning to validate this strategy for xenogeneic vaccines, which are not necessarily translatable to a clinical setting. It would be far more important to put the focus on epitopes that are entirely shared among mouse and human amino acidic sequences. In this way you validate the CD169 Ab in a system that incorporates the issue of breaking tolerance against self-melanoma antigens. Actually, the peptide Trp2(180-188) tested by the authors fulfills this criterion. Indeed, this peptide has been also described as an HLA-A2-restricted epitope, which is ideal to test in both wild-type H-2b and transgenic HLA-A2 mice.
As pointed out by the reviewer, the main novelty of the mouse in vivo part is the validation of our vaccination strategy with melanoma epitopes. We have previously shown efficient induction of T cell responses with a highly immunogenic model antigen (ovalbumin) and in this manuscript we show the activation of T cell responses towards multiple melanoma epitopes restricted by different MHC molecules. These melanoma epitopes represent tumor associated antigens for which tolerance exist. The epitopes that we selected have been previously described by others who also aimed to overcome tolerance (references 37-42). Our CD169 based vaccination strategy was successful in overcoming tolerance to several melanoma epitopes, including a Trp2 epitope that is conserved between mouse and human. We adjusted the discussion to emphasize the importance of overcoming tolerance against Trp2, as highlighted by the reviewer (see discussion page10 line 273).
To really validate anti-tumor responses generated by the anti CD169 antibody linked to melanoma peptides, it would be needed to test this against a melanoma tumor model, such as wild type and HLA-A2-expressing B16 melanoma in a therapeutic or at least preventive settings.
We have previously shown in a ovalbumin-expressing B16 melanoma model, that ovalbumin targeting to CD169 led to T cell responses that were able to suppress melanoma outgrowth in therapeutic settings. The melanoma epitopes utilized in this study have also been previously shown to suppress tumor outgrowth (reference 37-42) and our vaccination strategy induces similarly strong T cell responses. Although we agree with the reviewer that the CD169 targeting vaccination strategy should formally be tested in a melanoma setting, we believe that the real strength of this vaccination strategy will be in combination with tumor-specific antigens, such as neoantigens (as discussed at page 11 line 316). Unfortunately, to test tumor-specific neoantigen vaccination in a therapeutic anti-tumor setting was beyond the scope of this proof-of-principle study.
In addition, to this reviewer it is not clear what is the degree of novelty of figure 3. It seems that all the data presented have been previously shown. What is the contribution of this figure to the manuscript?
We understand the confusion after re-reading our text in which we refer to previous studies. In the study of Steiniger et al (reference 47) the presence of perifollicular macrophages that express CD169 is shown in human spleen. In the study by Pack et al. Immunlogy 2008, 123: 438)DEC205 staining is either combined with DC-SIGN or with CD169. To the best of our knowledge, our study is the first to show perifollicular cells co-expressing CD169 and DC-SIGN in human spleens. Park et al. (reference 63) showed co-staining of CD169 and DC-SIGN in human lymph nodes. We have adjusted the text to clarify this.
In the same line, there is no clear connection of the main topic of the manuscript with the analysis of the binding, uptake and internalization of the anti-CD169 Ab in human monocytes. The characterization of this antibody has been made after in vivo i.v. delivery, where it is uptaken by CD169 macrophages and then by DCs. However, it is not clear why is important to study this in human monocytes. It could be relevant to study this if the anti CD169 antibody was planned to be used in ex vivo monocyte-derived vaccination protocols. In that case, what would be the advantages of using this antibody?
Since it is not possible to isolate human splenic CD169+ macrophages and dendritic cells in sufficient numbers to do functional ex vivo assays, we are limited to in vitro studies to investigate the feasibility of antigen targeting to human CD169. As the human spleen stainings indicated co-expression of DC-SIGN and CD169, we decided to use INF-treated monocyte-derived dendritic cells that co-express these markers as an in vitro equivalent of the in vivo cell type.
Human monocyte-derived dendritic cells are frequently used as an in vitro culture system for the validation of DC receptor mediated uptake and antigen presentation to T cells. Thus far, to the best of our knowledge, it has not been shown that antigens targeted to CD169 using antibodies are taken up by monocyte-derived DCs and presented to T cells. Therefore, the data presented in figures 4 and 5 is novel and essential for further translation of our preclinical mouse studies. We do not aim to use the described strategy for monocyte-derived dendritic cell vaccination protocols, but we strongly believe that our studies are relevant for the development of a vaccine based on CD169 targeting in humans.

Reviewer 3 Report
The current paper is technically sound, nicely written, well-structured providing convincing evidences for the proof of concept. Further, the current work carried out in the manuscript is of clinical relevance. Minor comment In Figure 3, scale bars are absent from some panels. Only scale is mentioned but the bar is not shown. Eg. 3C and 3D respectively.
Author Response
The current paper is technically sound, nicely written, well-structured providing convincing evidences for the proof of concept. Further, the current work carried out in the manuscript is of clinical relevance. Minor comment In Figure 3, scale bars are absent from some panels. Only scale is mentioned but the bar is not shown. Eg. 3C and 3D respectively.
We thank the reviewer for the appreciation of our work. We have adjusted the scale bars in the figures.
Round 2
Reviewer 2 Report
I appreciate the efforts made by the authors to address the concerns raised by this reviewer. However, not many changes were included in the revised version of the manuscript. Overall, I consider these efforts insufficient.
First, I consider that immunizing with human-derived peptides to break tolerance against mouse-derived epitopes is not relevant for this translational setting, unless this is in the context of xenogeneic vaccinations, a concept that was nor introduced neither discussed. Hence, I would recommend to put the main focus of figure 2 on trp2 data. For instance, gp100 data could be included in a secondary position or even as supplementary data.
Second, given that the strength of the present manuscript is assessing the potential of anti-CD169 vaccination to break tolerance against melanoma-associated self-antigens. Thus, I consider necessary to formally test the anti-melanoma effects of anti-CD169 Ab immunization in vivo using a melanoma tumor model. Given the relevance of TRP2 epitope and the high responses achieved against it, this seems the most appropriate antigen. These data could be incorporated to figure 1. (I fully agree that neoantigen vaccination is out of the scope of this manuscript).
Although the rest of my concerns were well addressed in the Author response letter, some of these concerns, such as novelty of figure 3 and the relevance of using human monocytes, were not properly addressed in the manuscript.
In addition to my previous comments, now I realized that differences in IFN-gamma production in figure 5 are not statistically significant, which precludes authors from making strong conclusions. Given the importance of this assay for the message of the paper I would recommend to include more donors. These data could be part of figure 4 instead of a figure by itself.
As a minor comment, figure legends should be better structured, starting with the more general description, which is common for the different panels, and then describing the particular aspects shown. As an example, figure 1 should start with “Intravenous immunization with Ab:Ag conjugates in the presence of anti-CD40 Ab and Poly(I:C)…”
Author Response
I appreciate the efforts made by the authors to address the concerns raised by this reviewer. However, not many changes were included in the revised version of the manuscript. Overall, I consider these efforts insufficient.
We want to thank the reviewer for the concerns. We hope to address them in a satisfactory manner.
First, I consider that immunizing with human-derived peptides to break tolerance against mouse-derived epitopes is not relevant for this translational setting, unless this is in the context of xenogeneic vaccinations, a concept that was nor introduced neither discussed. Hence, I would recommend to put the main focus of figure 2 on trp2 data. For instance, gp100 data could be included in a secondary position or even as supplementary data.
We agree with the reviewer that the focus of figure 2 should be on the Trp2 data and that immunizing with human-derived peptides is less relevant in this context. We have adjusted the order of Figure 2 and the text in the results section (page 3-4).
Second, given that the strength of the present manuscript is assessing the potential of anti-CD169 vaccination to break tolerance against melanoma-associated self-antigens. Thus, I consider necessary to formally test the anti-melanoma effects of anti-CD169 Ab immunization in vivo using a melanoma tumor model. Given the relevance of TRP2 epitope and the high responses achieved against it, this seems the most appropriate antigen. These data could be incorporated to figure 1. (I fully agree that neoantigen vaccination is out of the scope of this manuscript).
To our opinion the main message of our manuscript is the efficacy of the in vivo CD169-targeting strategy for the induction of T cell responses against melanoma-associated antigens. We show in our manuscript that anti-CD169-targeting is as efficient as dendritic cell targeting using anti-DEC205. Whether or not tolerance can be broken will predominantly be dependent on the epitope and the T cell repertoire and less on the targeting strategy utilized. To test the CD169-targeting strategy we selected a number of melanoma antigens that had previously been shown to be able to break tolerance and induce T cell responses. While we agree with the reviewer that it would be of additional information to test the anti-CD169-Trp2 Ab immunization in a melanoma setting, we must point out that we have previously evaluated the efficacy of CD169 targeting in the B16OVA melanoma model (van Dinther et al. Frontiers in Immunology 2018). In this model both immunization with anti-CD169 conjugated to ovalbumin protein or SIINFEKL peptide exhibited strong anti-melanoma responses. Similar experiments with the CD169-Trp2 conjugates and B16F10 tumors would be a reflection of the capacity of Trp2-specific T cells to eliminate B16F10 tumors, since the efficacy of CD169-targeting to induce Trp2-specific T cells is clear. Therefore, we think that the additional value of these in vivo experiments is limited.
Although the rest of my concerns were well addressed in the Author response letter, some of these concerns, such as novelty of figure 3 and the relevance of using human monocytes, were not properly addressed in the manuscript.
We have better addressed the novelty of figure 3 and the relevance of using human monocyte-derived dendritic cells in the text (page 5-7 and 10).
In addition to my previous comments, now I realized that differences in IFN-gamma production in figure 5 are not statistically significant, which precludes authors from making strong conclusions. Given the importance of this assay for the message of the paper I would recommend to include more donors. These data could be part of figure 4 instead of a figure by itself.
The data in figure 5 are the combined data of five experiments with different MoDC donors and T cell batches and showed variation in the production of INFg by the T cells after antigen presentation by MoDCs. The technical replicates from four out of five donors were significantly different with a two-way ANOVA and Sidak’s multiple comparison correction (see figure in PDF). The data presented in Figure 5 are the means of each donor and were analyzed with a repeated measurements ANOVA with Sidak’s multiple comparison correction for the comparisons “control versus CD169” and “control versus DC-SIGN” are significantly different (0.034 and 0.0074 respectively). The comparison between CD169 and DC-SIGN was not significantly different. We adjusted the text accordingly (page 9).
As a minor comment, figure legends should be better structured, starting with the more general description, which is common for the different panels, and then describing the particular aspects shown. As an example, figure 1 should start with “Intravenous immunization with Ab:Ag conjugates in the presence of anti-CD40 Ab and Poly(I:C)…”
We have adapted all figure legends according to the suggestion of the reviewer.
